# WHAT PRESERVES THE EMERGENCE OF LANGUAGE?

## ABSTRACT

The emergence of language is a mystery. One dominant theory is that cooperation boosts language to emerge. However, as a means of giving out information, language seems not to be an evolutionarily stable strategy. To ensure the survival advantage of many competitors, animals are selfish in nature. From the perspective of Darwinian, if an individual can obtain a higher benefit by deceiving the other party, why not deceive? For those who are cheated, once bitten and twice shy, cooperation will no longer be a good option. As a result, motivation for communication, as well as the emergence of language would perish. Then, *what preserves the emergence of language?* We aim to answer this question in a brand new framework of agent community, reinforcement learning, and natural selection. Empirically, we reveal that lying indeed dispels cooperation. Even with individual resistance to lying behaviors, liars can easily defeat truth tellers and survive during natural selection. However, social resistance eventually constrains lying and makes the emergence of language possible.

## 1 INTRODUCTION

Unveiling the principles behind the emergence and evolution of language is attractive and appealing to all. It is believed that this research field is of great significance for promoting the development of enabling agents to evolve an efficient communication protocol (Nowak & Krakauer, 1999; Kottur et al., 2017; Chaabouni et al., 2019) or acquire existing one (Li & Bowling, 2019), especially when interacting with humans. Previously, many studies have investigated some intriguing properties of language and their effects on the emergence of language (Andreas & Klein, 2017; Lazaridou et al., 2018; Mordatch & Abbeel, 2018). The motivation behind these is that human language is considered as a remarkable degree of structure and complexity (Givon, 2013) and each character is the result of evolution, thus they believe that understanding the language itself is an indispensable step to take. Unlike existing work, we, from a different perspective, focus on a fundamental question that what made the emergence of language possible during evolution.

One of the dominant theories in the community of emergent communication is: cooperation boosts language to emerge (Nowak & Krakauer, 1999; Cao et al., 2018). Hence, there has been a surge of work investigating this field in cooperative multi-agent (mostly two agents) referential games (Lazaridou & Peysakhovich, 2017; Kottur et al., 2017; Das et al., 2017; Evtimova et al., 2018; Lazaridou et al., 2018), a variant of the Lewis signaling game (David, 1969). However, they seem to miss some basic elements in the human language. On one hand, human language emerges from the community, not just two persons, after all, language is learnable and can spread from one place to other (Dagan et al., 2020). Studying a language in two-player games is like looking at the world through a keyhole. On the other hand, many works make an agreement that prior to the emergence of language some pre-adaptations occurred in the hominid lineage, and one of the candidates is the ability to use symbols (Deacon, 2003; Davidson, 2003; Christiansen & Kirby, 2003). It seems understanding the emergence of symbolic signals is the key to approach the truth of the origin of language (Deacon, 1998). However, chimpanzees have demonstrated a degree of language capacity by using arbitrary symbols as well as the ability for the cross-modal association, abstract thought, and displacement of thought in time (Meddin, 1979). So why don't they have a language like us? One of the theory is selfishness has kept animal communication at a minimum (Ulbaek, 1998). In more detail, if an individual can obtain a higher benefit by deceiving the other party in the cooperation, why not deceive? Once deception emerges, mistrust among individuals will haunt. For those who are cheated, once bitten and twice shy, cooperation will no longer be a good option. As a result,

motivation for communication, as well as demands of the emergence of language will perish. But human beings are so special since we have overcome this kind of obstacle and evolved language. Then, *what preserves the emergence of language?* We aim to answer this question in a brand new framework of agent community, reinforcement learning (RL), and natural selection. We believe this process should occur in the pre-language period since lying is possible as long as agents can communicate. Therefore, our investigating communication protocol uses symbols to transmit meaning based on a social convention or implicit agreement.

In this paper, we introduce several agents to form one community and allow natural evolution and elimination among them. Both liars (agents tell lies) and truth tellers (agents always tell truth) exist in the community. Each tournament, every agent is supposed to play a non-cooperative game (Nash Jr, 1950; Nash, 1951; Schelling, 1958; Binmore et al., 1986; Von Neumann & Morgenstern, 2007) with others. In our multi-round bargaining game, agents are required to reach an agreement about how many items to give out so that the total quantity can satisfy the market's demand and they can keep their loss to the minimum in the meantime. We believe this is a perfect fit for the nature of human beings and more common in the hominid lineage compared with the cooperation game. Importantly, during the process of natural selection, the fraction of liars and truth tellers may change from time to time and this allows us to observe what factor imposes influence on the motivation of communication, which is the prerequisite of the emergence of language. It is worthy of note that pre-language communication was subject to the constraints of Darwinian evolution. While linguistic change, which began in the post-language communicative era of hominid evolution, is by and large tied to society and culture (Givón & Malle, 2002; Li & Hombert, 2002). Thus, we disregard the factors related to linguistic change since we are investigating motivation for communication from which language evolved.

Moreover, apart from the normal setting mentioned above, we add up two more rules to further dig out. Firstly, we introduce a credit mechanism for truth tellers. In other words, we make sure truth tellers know the existence of liars which is one step in the evolution process. Specifically, every liar has credit in the mind of truth teller, and the credit varies with the profit of truth teller. Cooperation would be impossible between two agents as soon as the credit drops to negative. Secondly, an additional penalty will be brought in as a price of lying, and we consider it as social pressure for resisting lying behaviors. All in all, we want to make a thorough investigation about how the individual or social resistance to lying affects communication.

Empirically, we show that in normal settings, two truth tellers can make a fair agreement, and liars can achieve a huge advantage over truth teller by telling lies. As for two liars, there is always a better liar that gains relative more than the other. In the credit setting, liars can learn a sophisticated lying strategy that deceives the credit mechanism and makes more profits meanwhile. In both settings, as time goes on, truth tellers seem not to show enough competition against liars and thus die out. In the society setting, we find out liars are afraid of lying if punishment is sufficiently large. This again proves the theory (Ulbaek, 1998): in the human lineage, social cooperation based on obligatory reciprocal altruism (Trivers, 1971) as well as a system which punishes people morally and physically for cheating has evolved. In such an environment language is finally possible.

## 2 EXPERIMENTAL FRAMEWORK

### 2.1 GAME SETTINGS

We explore emergent language in the context of multi-round non-cooperative bargaining game (Nash Jr, 1950; Nash, 1951) as illustrated in Figure 1. The core is binding cooperative strategy with the highest profit is impossible whereas selfish strategy sometimes can. In this case, the behavior of telling lies can be meaningful since it undermines cooperation and grabs more benefits from others.

In the game, two agents $i, j$ bargain over how to satisfy the demand of market. Agents are presented with $N$ different items. They possess a fixed number of quantity for each item ($\{q_n^i\}_{n=1}^N, \{q_n^j\}_{n=1}^N$), and have their own hidden utilities for $N$ items ($\{u_n^i\}_{n=1}^N, \{u_n^j\}_{n=1}^N$). Agents move sequentially. Suppose at bargaining round $t$, it is turn for agent $i$ to be proposer and it makes proposal $\{p_{t,n}^i\}_{n=1}^N$ about how many to give out to the market, which means the other agent $j$ should contribute the rest $\{d_n - p_{t,n}^i\}_{n=1}^N$ to the market, where $d_n$ is the market demand for item $n$. Then agent $j$ would

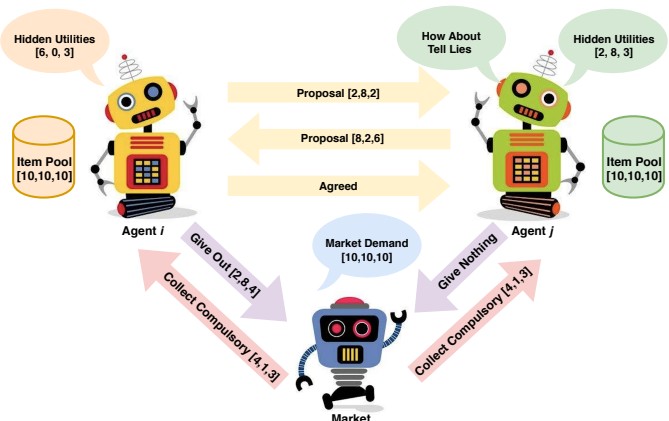

Figure 1: Overview of the bargaining game. Two agents learn to bargain over how to satisfy the demand of market. The reward of each agent is the value of remaining items. If the demand of market is not satisfied, the difference will be collected from both agents evenly in a mandatory way.

choose to accept the proposal or not. The game will be terminated when they make an agreement about the proposal or the number of bargaining rounds reaches the upper limit $T_{\max}$, and agents $i$ and $j$ will receive rewards $\sum_{n=1}^{N}(q_n^i - p_{t,n}^i) \times u_n^i$ and $\sum_{n=1}^{N}(q_n^j - d_n + p_{t,n}^i) \times u_n^j$ respectively or get zero when no agreement has been made in the end. In order to make the reward comparable, it will be further normalized by the maximum reward achievable for each agent. Both agents want to keep as more items as possible rather than giving out. Therefore, agents are supposed to seek a tradeoff between keeping more items and satisfying the market demand.

In this part, we illustrate how lying mechanism works, where lying means the proposal will not be followed by actions. Suppose agents $i, j$ satisfy the market demand at round $t$, and agent $j$ chooses to tell lies about the proposal and then gives nothing to the market. This leads to that the market will have demand gap which is $\{p_{t,n}^j\}_{n=1}^N$, and the market would force each agent to hand over half of the gap for remedy. It is noted that liars are allowed to tell lies about any items. In our settings, we conform to the principle of high risk and high return. To illustrate, when $d_1 = 10$ agent $j$ tells lies and keeps offering $p_{t,1}^j = 9$, agent $i$ is definitely delightful and more likely to take $d_1 - p_{t,1}^j = 1$. Although agent $j$ can easily put others on the hook, it cannot gain a large advantage since the final result is not much better than evenly contributing. On the contrary, if agent $j$ takes $p_{t,1}^j = 1$, it allows agent $j$ keeping more items and obtaining more profits. However, this proposal is less appealing to agent $i$. The key point is we want to link attractive lies to relatively lower profits.

As agents take turns to play, first-mover has absolute advantages since the other one is compelled to accept unfair proposal or get nothing. We once tried to convert our bargaining game from moving alternately to moving simultaneously to solve this problem. In detail, two agents make proposals at the same time and their total quantity should surpass the demand. However, it turns out that agents only learn to evenly divide the market demand. In this setting, no agent can guarantee an agreement, therefore the proposal made by an agent could be more conservative, meaning it is hard for the agent to infer others' utilities during bargaining. Ultimately, we adopt the strategy (Cao et al., 2018) by randomly sampling $T_{\max}$ between 4 and 10 to mitigate the first-mover effect.

## 2.2 GAME RULES

**Natural Selection.** We have claimed that it is more reasonable to study the emergence of language in a community since natural selection played an essential role in the acquisition of a new ability. Twins, for instance, sometimes can develop cryptophasia, which is a language no one can understand except the twins. However, such a language is an incomplete work and will be eliminated by nature. In the community, there are $M$ agents: some of them are liars and the rest are truth tellers. In each tournament, each agent will play the game with all other agents in the community. Natural selection occurs after every $K$ tournaments. To be specific, natural elimination is reflected by that the agent with the lowest profit will be weeded out from the community. As for genetic evolution, a new agent that evolved from the elite will fill the opening. The new agent will be trained to learn its own policy

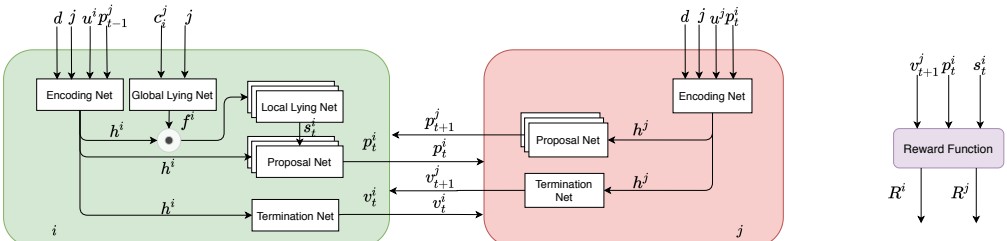

Figure 2: Network architecture.

by playing with the remaining agents, but it is initialized by the parameters of the agent with the highest profit. At the meantime, the remaining agents are also allowed to be trained for the purpose of adapting the new agent.

**Credit Mechanism.** Every truth teller has a credit record for each liar. At the beginning, truth tellers consider all other agents are trustful, and credit records are initialized as zero. The credit record will be altered every time after a truth teller plays with a liar. To illustrate, truth teller $j$ plays a game with liar $i$ and gets reward $r^j$, then the credit record $c_i^j$ for $i$ will add a value of $1 - \exp(0.5 - r^j)$. In this setting, a truth teller cannot tolerate that the profit is less than half of the maximum, and it will realize that it might be cheated since cooperation can definitely achieve more. Also, it is much easier for a liar to lose trust than acquiring it. We want to reflect the reality that one lie can destroy a person no matter how much truth he told before. In addition, the liar can observe its credit records from truth tellers when making decisions.

### 2.3 NETWORK ARCHITECTURE AND TRAINING

At each round $t$, suppose it is the turn for agent $i$ to make a proposal, first, agent $i$ obtains a partial observation $o_t^i$, which consists of four parts: market demand $d$, its own utility $u^i$, proposal $p_{t-1}^j$ made in the previous round $t-1$ by its opponent $j$, and ID of $j$. An encoding network $e^i(\cdot)$ which is realized by a two-layer feedforward neural network, maps the observation $o_t^i$ to an encoded vector $h_t^i$. Then, a termination network $\pi_{\text{term}}^i(\cdot)$, which is realized by a single-layer feedforward neural network, takes $h_t^i$ as input and outputs a binary action $v_t^i$ that determines whether agent $i$ accepts the proposal $p_{t-1}^j$. If it accepts, the game ends and both agents receive their rewards based on the agreement; otherwise, agent $i$ needs to give an alternative proposal. Truth teller and liar are different in generating a proposal. If agent $i$ is a liar, it sends $h_t^i$ into a local lying network $\pi_{\text{lie}}^i(\cdot)$ to output $s_t^i$ that indicates whether to tell lies for each item. Currently, the local lying network has a separate two-layer feedforward neural network for each item, however, it can also be instantiated by a single network that outputs decisions for all the items. Next, agent $i$ feeds $h_t^i$ and $s_t^i$ into a proposal network $\pi_{\text{prop}}^i(\cdot)$ to produce the proposal $p_t^i$. The proposal network also has a separate feedforward neural network for each item $n$, which outputs a distribution over $\{0, ..., q_n^i\}$. The proposal is sampled from the distributions. If agent $j$ is truth teller, $s_t^i$ is set to 0 by default. After that, it is the turn for agent $j$ to move and it will repeat the procedure. Note that when the game ends, the liar determines whether to make deception about each item according to the output of its local lying network. The network architecture is illustrated in Figure 2.

When introducing credit mechanism, liar $i$ additionally has a global lying network $\pi_{\text{LIE}}^i$ that takes inputs as credit records $c_i^j$ and ID of its opponent $j$. The global lying network acts like a gate for local lying network $\pi_{\text{lie}}^i$ and outputs a binary variable $f^i$ at the beginning of each game that determines whether to tell lies at the subsequent bargaining round. Unlike $\pi_{\text{lie}}^i$ that controls the agent at each round and focuses on how to achieve more profits in just one game, $\pi_{\text{LIE}}^i$ takes the whole picture and aims to deceive the credit mechanism of truth tellers in order to get more profits in a long run. It considers questions as follow: what would happen if it lies in one game? would others realize that it is a liar?

During training, there are three kinds of policies, $\pi_{\text{term}}$, $\pi_{\text{prop}}$, and $\pi_{\text{lie}}$ (if existed) to be updated for each agent, which are parameterized by $\theta_{\pi_{\text{term}}}$, $\theta_{\text{prop}}$, and $\theta_{\text{lie}}$, respectively. For each game, the policies are updated towards maximizing its own expected reward. Suppose agent $i$ plays with agent

$j$, the objective of agent $i$ is

$$\mathcal{J}(\boldsymbol{\theta}_{\boldsymbol{\pi}^i}) = \mathbb{E}_{\tau \sim \boldsymbol{\pi}^i, \boldsymbol{\pi}^j}[R_j^i(\tau)], \qquad (1)$$

where $\boldsymbol{\theta}_{\boldsymbol{\pi}^i} = \{\theta_{\pi_{\text{term}}^i}, \theta_{\pi_{\text{prop}}^i}, \theta_{\pi_{\text{lie}}^i}\}$, $\boldsymbol{\pi}^i = \{\pi_{\text{term}}^i, \pi_{\text{prop}}^i, \pi_{\text{lie}}^i\}$, and $R_j^i(\tau)$ is the reward that agent $i$ receives from trajectory $\tau$, playing with agent $j$. The trajectory $\tau$ of one bargaining game is defined as $\{o_t, \boldsymbol{a_t} = \{v_t, s_t, p_t\}, r_t\}_{t=1}^T$. Then, the gradient of the policies is computed by REINFORCE (Williams, 1992) and can be further derived as

$$\nabla_{\boldsymbol{\theta}_{\boldsymbol{\pi}^i}} \mathcal{J}(\boldsymbol{\theta}_{\boldsymbol{\pi}^i}) = \mathbb{E}_{\tau \sim \boldsymbol{\pi}^i, \boldsymbol{\pi}^j}[R_j^i(\tau) \cdot \sum_{t=1}^T (\nabla_{\theta_{\pi_{\text{term}}^i}} \log \pi_{\text{term}}^i(v_t^i|o_t^i)$$
$$+ \nabla_{\theta_{\pi_{\text{lie}}^i}} \log \pi_{\text{lie}}^i(s_t^i|o_t^i) + \nabla_{\theta_{\pi_{\text{prop}}^i}} \log \pi_{\text{prop}}^i(p_t^i|o_t^i, s_t^i)) + \boldsymbol{\lambda} \nabla_{\boldsymbol{\theta}_{\boldsymbol{\pi}^i}} \mathcal{H}(\boldsymbol{\pi}^i)], \qquad (2)$$

where $\boldsymbol{\lambda}$ are hyper-parameters and $\mathcal{H}$ is the entropy regularizer to encourage exploration. As for $\pi_{\text{LIE}}^i$, the policy gradient is computed as

$$\nabla_{\theta_{\pi_{\text{LIE}}^i}} \mathcal{J}(\theta_{\pi_{\text{LIE}}^i}) = \mathbb{E}_{\pi_{\text{LIE}}^i}[\sum_{k=1}^K \sum_{j=1, j \neq i}^M G_{j,k}^i \cdot \nabla_{\theta_{\pi_{\text{LIE}}^i}} \log \pi_{\text{LIE}}^i(f_k^i|c_{i,k}^j, j)], \qquad (3)$$

where $G_{j,k}^i$ is the return of agent $i$ playing with agent $j$ started from tournament $k$ after previous natural selection and $G_{j,k}^i = \sum_{l=k}^K R_{j,l}^i$.

## 3 EXPERIMENTS

### 3.1 CAN LANGUAGE SURVIVE WITH NO RESTRICTIONS IN ONE COMMUNITY?

**Experimental setting.** In this experiment, we are investigating what would happen when there is nothing to counter lying behaviors in the community. There are 8 agents living in a community and trying to make profits by playing bargaining games with others. Half of them are truth tellers and they do not have credit records for liars, which means they are unable to make any response to lying behaviors. The rest are liars which possess local lying network but not the global one. An agent has to play a game with everyone else in each tournament. Every 10 tournaments, the community is updated according to natural selection. We have two training phases. First we train 4 truth tellers for 200k episodes. In each episode, two random truth tellers are sampled to play the game. After all truth tellers have learned how to infer intention of others and make proper proposals, we begin to train liars for 200k episodes by sampling two random agents from the community to play the game in each episode. Note that at least one sampled agent is a liar and the models of truth tellers would be frozen in this phase. Each episode corresponds to a batch of 128 games.

For game settings, we have $N = 3$ different items. Market demand $\{d_n\}_{n=1}^N$ and items in possession $\{q_n\}_{n=1}^N$ are set to 10. Utilities $\{u_n\}_{n=1}^N$ are sampled uniformly between 0 and 5 for each agent. For each game, $T_{\max}$ is uniformly sampled from 4 to 10.

**Results and analysis.** Figure 3a shows the learning curves of all truth tellers in terms of reward in the first training phase. And we can find out 4 truth tellers converge to a similar level (around 0.64), demonstrating that they have learnt to make a fair proposal that can be accepted by both party. To some extent, their proposal shows cooperation and is a win-win strategy since they can obtain more

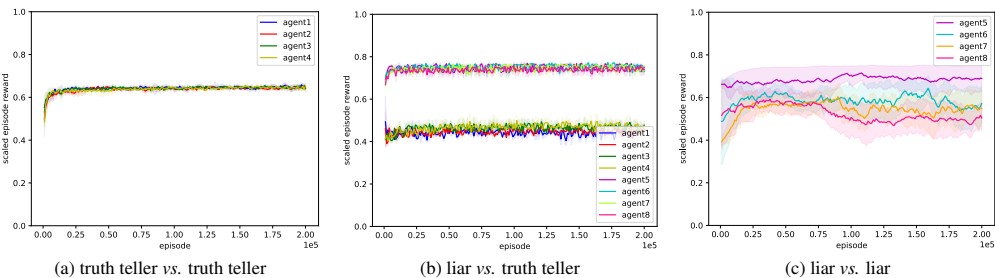

(a) truth teller *vs.* truth teller      (b) liar *vs.* truth teller      (c) liar *vs.* liar

Figure 3: Learning curves in terms of rewards in different games.

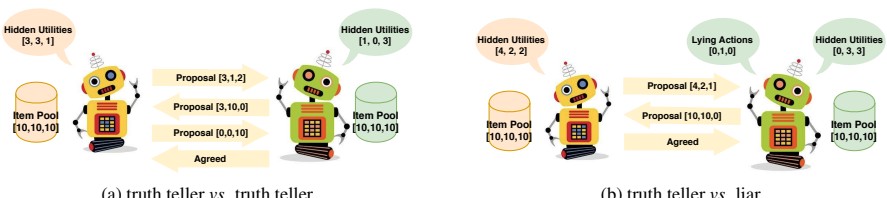

(a) truth teller *vs.* truth teller        (b) truth teller *vs.* liar

Figure 4: Examples of bargaining between agents

than $0.5$ (*i.e.*, the profit achieved by dividing items equally). Figure 3b and 3c show the learning curves of all agents in terms of reward in the second training phase. Figure 3b presents the games between liar and truth teller, and Figure 3c presents the games between liars. From Figure 3b, 4 liars perform better in the game than truth tellers and reach above $0.75$, while truth tellers only get around $0.45$ that is lower than the equally dividing strategy. It shows lying behaviors indeed help liars to take a dominant position over truth tellers. The performance of liars showed in Figure 3c is between the win-win strategy and equally dividing strategy. It turns out there exits a liar that outperforms all other liars when it comes to liar *vs.* liar games. Figure 4 visualizes two examples of bargaining using learned policies between truth tellers and between truth teller and liar.

We further analyze the change of the community under natural selection. Figure 5 shows as natural selection goes, truth tellers are gradually wiped out from the community and eventually liars take over the whole community. Liars are able to obtain more profits by bargaining with truth tellers so that they can fill the reward gap caused by liar *vs.* liar games. When liars are the only kind in the community, their average rewards begin to decline since they are unable to squeeze the profit from truth tellers. The results suggest cooperation is finally impossible as long as lying behavior is possible in the community. Liars would never try to cooperate with others, and

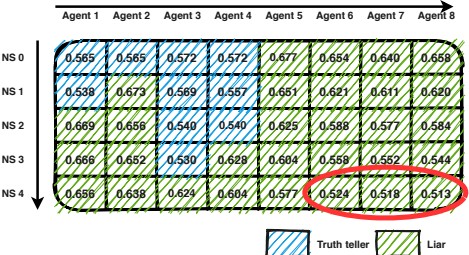

Figure 5: Evolution of the community. The number is the average reward during a natural selection period.

they undermine the motivation of communication (lower and lower rewards). As many hypotheses state cooperation is a prerequisite for the emergence of language (Cao et al., 2018; Nowak & Krakauer, 1999), we believe more sophisticated language is hard to evolve in such a community. Our finding also corroborates the hypothesis that if it is better to lie than to tell truth, why is all this elaborate coding of thoughts into speech necessary whereas the effective strategy is just not to listen (Ulbaek, 1998), as evidenced by that some liars get nearly the same as equally dividing and hence communication is not necessary, not to mention it incurs extra cost in reality.

## 3.2 HOW INDIVIDUAL RESISTANCE TO LYING AFFECTS THE EMERGENCE OF LANGUAGE?

**Experimental setting.** In the second experiment, we turn to the scenario where truth tellers would counter lying behaviors on their own by refusing to bargain with agents they consider as liars. We are investigating that whether individual countermeasure is able to suppress lying behaviors and is a necessity for language to emerge. In this settings, we still have 4 truth tellers and 4 liars in the community. For truth tellers, they have credit records for all the liars. They would not play following games with those whose credit is negative until next natural selection. And both parties get reward 0 for the games where one party is absent. Credit mechanism is designed to help truth tellers to discern liars and punish them within their own ability. However, it is still possible for one agent to get low reward even if no one lies due to the random utilities. To reduce such re-

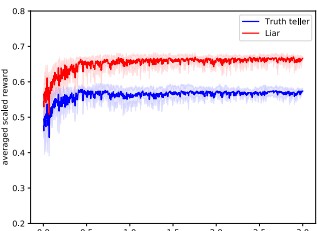

Figure 6: Learning curves in terms of rewards in games of liar *vs.* truth teller.

ward variance, truth tellers update credit records based on a batch of games (the size is 16). For liars, they have the global lying network apart from local one, which gives them opportunities of seeking strategies to deceive credit mechanism and keep their advantages. In addition, the global lying

network is trained for 3k episodes with batch size of 16. As for other networks, we use pre-trained models from the first experiment. The game settings are the same with the first experiment.

**Results and analysis.** Figure 6 illustrates an example of learning curves of liar *vs.* truth teller in terms of average reward in one natural selection period. We can see that the liar converges to about 0.67 and the truth teller is able to obtain an above-average reward of 0.57. Comparing to the first experiment, the difference of rewards between truth teller and liar narrows down. This is the consequence of the compromise between the credit mechanism and global lying network. It turns out the global lying network assists the liar to deceive the credit mechanism in some way since it can achieve more than that it always tells truth (around 0.64). However, the liar still needs to sacrifice some benefits in order to gain enough trusts from its opponent. As observed in the experiment, the liar has learned an effective strategy to accomplish the mission. Figure 7 shows the lying frequency in each tournament during the natural selection period. We can find out that in the first 4 tournaments, the lying frequency is relatively

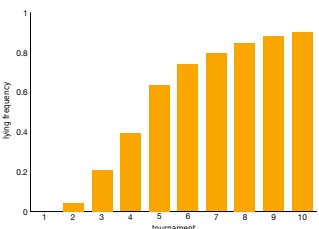

Figure 7: Lying frequency.

low since liars need to earn trust so that truth tellers would not identity them and refuse to play games at an early stage. With the accumulation of enough trust, liars start to tell lies more often to make more profits. As the natural selection approaches, there is less and less need to pretend for liars since the punishment imposed by truth tellers is meaningless at the end of natural selection period. This explains that the lying frequency gradually increases and is relatively high after 4 tournaments.

The competition becomes more fierce since the gap of average rewards between agents is closer than in the previous experiment. However, similar phenomenon in natural selection can be observed in Figure 8, *i.e.*, liars gradually taking over the community, though the process is slower than the previous experiment. The process is slower, because the truth teller is not always ranked at the bottom of the list in terms of rewards. However, as weaker liars are replaced by the elite liar by natural selection, truth tellers progressively decline to the bottom and are eventually wiped out from the community. Once all agents in the community are liars, similar conclusion for the emergence of language can be drawn as the first experiment. In addition, it is indicated that the credit mechanism is effective to protect truth tellers

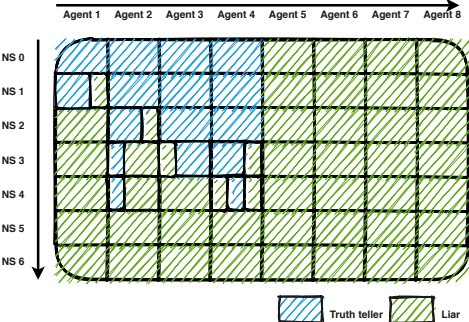

Figure 8: Evolution of the community with individual resistance to lying behaviors, where each grid is formed by the results of three runs and hence fraction appears.

to some extent, but liars can still find their way to overcome it. Furthermore, individuals are unable to impose enough force to suppress lying behaviors. Therefore, we claim that it is not easy to resist lying behaviors relying solely on individuals and more compelling force is needed to create a harmonious cooperative environment.

### 3.3 How Social Resistance to Lying Affects the Emergence of Language?

**Experimental setting.** In the final experiment, we seek for a more severe punishment to resist lying behaviors. It reminds us that social organization, *e.g.*, enterprise or government, would penalize anyone who breaks rules, *e.g.*, breach of contract, by imposing a fine or putting into prison. Inspire by this, we introduce two kinds of artificial pressure on liars to counterbalance their lying behaviors, mimicking two types of aforementioned punishment. One is designed for local lying mechanism, and a penalty $\alpha \times |\text{lies}|$ is directly added to reward function to disfavour lying behaviors. $\alpha$ is a hyperparameter which is considered as the punishment strength in reality and $|\text{lies}|$ is the number of items a liar tells lies about in one game. Another is for global lying mechanism, every time the gap of rewards between two agents exceeds over a threshold (*i.e.*, 0.2),

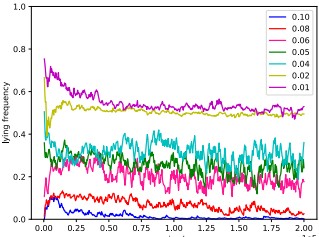

Figure 9: Learning curves in terms of lying frequency of different $\alpha$.

the system believes lying behavior has occurred and the liar will be banned for next $L_{\text{prison}}$ games

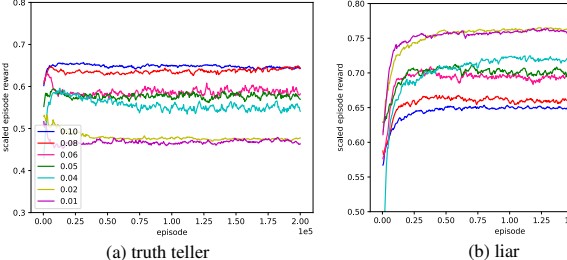

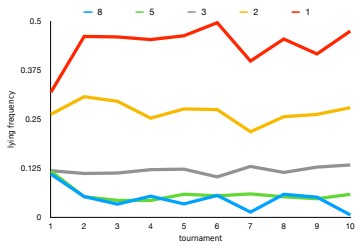

(a) truth teller                    (b) liar

Figure 10: Learning curves of truth teller and liar in terms of rewards with different $\alpha$.

Figure 11: Lying frequency of different $L_{\text{prison}}$.

with the same player as punishment. However, in order not to damage the interest of truth tellers, the system will enforce equally dividing for canceled games and both sides get reward $0.5$ instead. As for training procedure and game settings, they are the same as the first experiment.

**Results and analysis.** Figure 9 shows the learning curves of different $\alpha$ in terms of lying frequency. We find out that when $\alpha$ is under the threshold of $0.02$, it has little influence on resisting lying behaviors. When the value of $\alpha$ exceeds $0.02$ and continues to increase (*i.e.*, more and more fines), the lying frequency gradually decreases to zero. And the rewards of both truth tellers and liars converges to a similar level, showing they begin to cooperate, as illustrated in Figure 10. In addition, similar pattern is also observed in Figure 11. If the prison time $L_{\text{prison}}$ is long enough, the global lying network tends to always output zero. It shows liars would turn into truth tellers as long as the system exerts sufficiently severe punishment.

In this experiment, our results show that artificial penalties inspired by existing social rules can restrain lying behaviors fundamentally compared with individual forces. With the help of strict social pressure, agents in the community are willing to cooperate with each other honestly. This is the premise of hypothesis that cooperation boosts language to emerge (Nowak & Krakauer, 1999; Cao et al., 2018). Only under such a environment, language is finally possible.

## 4 DISCUSSION

We found that language is impossible to be preserved when some agents are entitled to lying ability in a community with natural selection. When two agents solve a simple non-cooperative task in a bargaining way, agents which are not prepared for lying behaviors easily fall into a very unfavorable situation. And most of the agents in the community adapt to lying during a long evolutionary process based on our simulations, leading to that cooperation is no longer a good option and there is no intention to communicate. This motivates us to unearth what preserves language from lying behaviors. According to existing human social behaviors and rules, we design the credit mechanism and artificial penalties, reflecting individual and social resistance to lying behaviors, in reality, to investigate how they affect the emergence of language. Follow-up experiments suggest the limitations of individual resistance. Individuals are unable to obtain full information to determine whether their opponents are lying and hence this is leading to more conservative sanctions that do not effectively suppress lying behaviors. On the contrary, the social system has the right to access all. Thus, it can impose more accurate and severe punishment directly to lying behavior and retain a system with obligatory reciprocal altruism, which provides the soil for the evolution of language. Based on the results above, we hypothesize that social resistance is the key to preserving the language other than individual resistance. Besides, each agent is realized by neural networks. The results are stable across hyper-parameters and different kinds of bargaining games since we also try settings in (Cao et al., 2018).

From the perspective of artificial intelligence, our results stress the importance of controlling the external environment of language emergence. It is well known that enabling intelligent agents to communicate with humans is always one of the ultimate goals for the AI community. If we want to achieve this, it is not enough just to teach agents about human language. Unlike the unemotional machine, the human can convey meaning different from the literal one, reflected in acts such as lying and playing jokes. We want to emphasize that it is impossible to evolve language if agents are unaware of such behaviors of human beings and existing language can be vulnerable to facing them.

Creating an environment conducive to cooperation seems particularly significant. Also, we present two proof-of-concept examples of how to maintain a favorable environment for the emergence of language by directly penalizing lying behaviors in reward functions or banning liars from games. In future work, we would look for less *ad hoc* ways for providing good soil for language to thrive and investigate how agents develop such resisting mechanism naturally in a more precise way.

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

## A  IMPLEMENTATION DETAILS

Embedding sizes, as well as all neural network hidden states, have dimension 100. And we use ADAM optimizer (Kingma & Ba, 2014) with default settings to optimize the model of each agent. For entropy regularisation hyperparameters $\boldsymbol{\lambda}$, we have different value for different policies, *i.e.*,, 0.05 for $\pi_{\text{term}}$, 0.005 for $\pi_{\text{prop}}$ and 0.05 for $\pi_{\text{lie}}$. In addition, during two phase training, we use an exponentially smoothed mean baseline for reducing reward variance as (Cao et al., 2018), which is defined as $b_{new} = 0.7 \cdot b_{old} + 0.3 \cdot R$. As for the training phase of genetic evolution, we train agents with 200k episodes. In more details, if the new agent is evolved from truth teller, training happens in all scenarios randomly. Otherwise, training happens in liar *vs.* liar and truth teller *vs.* liar. And agents are selected randomly according to different scenarios.

An encoding network $e_i(\cdot)$ is realized by a two-layer feedforward neural network. Termination network $\pi_{\text{term}}(\cdot)$ is realized by a single-layer feedforward neural network followed by a sigmoid function. The local lying network $\pi_{\text{lie}}(\cdot)$ has a separate two-layer feedforward neural network for each item and each is followed by a sigmoid function, however, it can also be instantiated by a single network that outputs decisions for all the items. And the global lying network $\pi_{\text{LIE}}(\cdot)$ also has two-layer feedforward neural network followed by a sigmoid function. As for $\pi_{\text{prop}}$, it is parametrised by 3 separate single-layer feedforward neural networks followed by a softmax function.

