# OpenReview forum: "What Preserves the Emergence of Language?"
_ICLR.cc/2021/Conference — Reject_

### Official Review · AnonReviewer4 · 2020-10-20
**This work investigates conditions under which communities of cooperative agents are stable. While the experiments are interesting, there's a lack of motivation behind fundamental assumptions made in this work**

**Rating:** 3
**Confidence:** 5

**Review:**

##### Summary #####
This paper investigates conditions under which communities of cooperative agents are stable. Communities in multi-round bargaining games with evolutionary dynamics are evaluated in three main setups. The first imposes no restrictions on the agents' behavior and is shown to be easily invaded by deceitful agents. The second enables agents to refuse to bargain with deceitful agents. Nevertheless, such communities are shown to be invadable. Finally, in the third setup, a global punishment system is shown to be able to drive out deceitful invaders. The main take-home message is that, when lying is an option, agents(' communities) need to be prepared for it.

##### Reasons for score #####
I vote for rejecting this submission. The main reason, further detailed in "Cons" below, is that I am not convinced that the problem this manuscript addresses is not an artifact of the rather strong assumption of selfish individuals and communities driven solely by functional pressure. I agree with the authors that we need to start looking at more naturalistic setups (e.g., communities instead of two-player games). However, this also relates to the evolutionary dynamics we take into consideration, and to a careful motivation of the setups we analyze.

##### Pros #####

+ Language emergence is a topic that has drawn renewed interest from multiple disciplines. Studying the conditions under which agents' acquired behaviors are (not) stable in a dynamic setting is important to understand their underpinning.
+ The setups are well explained and easy to follow

##### Cons ######
- Much of the motivation of this paper relies on the argument that there are strategies that functionally dominate cooperation. However, the view that functional pressure is all there is to evolution is rather outdated. If the authors wish to stick to this line of argumentation, I encourage them to give strong arguments for why we should disregard other factors, such as transmittability/acquisition (e.g., Kirby et al. 2015, "Compression and Communication in the Cultural Evolution of Linguistic Structure", or Brochhagen et al. 2018, "Coevolution of Lexical Meaning and Pragmatic Use") or neutral models (e.g., Reali & Griffiths 2011, "Words as alleles: connecting language evolution with Bayesian learners to models of genetic drift.", or Perfors & Navarro 2014, "Language Evolution Can Be Shaped by the Structure of the World"). Along the same lines, the view that functional pressure needs to apply to only individuals, and that these are fully selfish, also requires evidence or needs to be clearly marked as a rather strong assumption. The claim that "[...] in order to ensure the survival advantage of many competitors, animals are selfish in nature" flies in the face of the whole line of research on animal alarm calls (e.g., Zuberbuehler 2009, "Survivor Signals: The Biology and Psychology of Animal Alarm Calling"); and, for that matter, the existence of any social animal. In sum, as it stands, I am not convinced that the problem this manuscript addresses is not an artifact of the assumption of selfish individuals and communities driven solely by functional pressure.
- Terminology is very sloppy at times. What is "successful evolution" (p. 1) or "naive natural selection" (p.8)?
- Writing could be greatly improved. There are a lot of typos and borderline grammatical sentences
- References are wrong (e.g., Lewis' work on convention is not from 2008 but from 1969. The same goes for all the references on seminal work in game theory)
- Many assumptions in the setup are not motivated. For instance, is there a reason to use a discrete dynamic over a continous one? (p. 3) Why use a multi-round bargaining game to model language evolution by contrast to, say, an established signaling game-style setup with conflict of interests? How is the "credit mechanism" and the "market" motivated/grounded in light of this being an investigation about language emergence? (p. 3)
- The lines in Figure 3a and Figure 3b are very hard/impossible to read. Particularly Figure 3b.
- The y-axis in the plots in Figure 3 are not aligned, rendering a comparison across conditions hard (and visually misleading)

---

> ### Author Response · Authors · 2020-11-15
> **Reply to AnonReviewer 4**
>
> \>>> Much of the motivation of this paper relies on the argument that there are strategies that functionally dominate cooperation.
>
> Transmittability, compositionality, learnability, grammatical and semantic categories, and many other properties make the human language unique among natural communication systems. However, human languages, like species, evolve over time. As a complex communication system, language should evolve from an immature communication system at an early stage. This is the period we want to investigate when naïve communication protocol, not necessarily possessing the properties owned by language (e.g. cryptophasia created by twins is not culturally transmitted), was just created and used by agents in a community. The main reason is that lying behavior should emerge once communication is feasible since some species (e.g., vervet monkey) have proved to express lying behavior in their daily life. Thus, we disregard other factors, such as transmissibility, since these properties do not contradict the conclusion we obtained. As for neutral models, it is defined to indicate how languages can change at the level of linguistic variants (Reali & Griffiths, 2011. Words as alleles: connecting language evolution with Bayesian learners to models of genetic drift). However, we are investigating the motivation of communication, which is the prerequisite of further evolution. As for motivation, we need to figure out whether communication can bring enough benefits since communication itself incurs additional resources. We are not investigating the distribution of word frequencies or the relationship between word frequencies and other factors.
>
> \>>> The claim that "[...] in order to ensure the survival advantage of many competitors, animals are selfish in nature" flies in the face of the whole line of research on animal alarm calls.
>
> First of all, as illustrated in (Zuberbuehler 2009, "Survivor Signals: The Biology and Psychology of Animal Alarm Calling"), alarm call is beneficial to a signaler if it increases the reproductive success since receivers prefer individuals as mating partners that are more willing to produce risky alarm calls in the presence of predators. Also, alarm signal is possible to trigger a subsequent group effort to get rid of the predator, then this behavior is profitable since it decreases the vulnerability of the signaler. All in all, parts of hypotheses about the presence of alarm calls are based on the assumption it is beneficial to the signaler, not to others. Therefore, our argument indeed does not fly in the face of the whole line of research on animal alarm calls since animal may do it for its own survival. Moreover, the existence of social animals does not mean animals are selfless facing the conflict of interests.
>
> \>>> Is there a reason to use a discrete dynamic over a continuous one?
>
> We use discrete symbols over continuous ones to communicate since it can be easily interpreted and analyzed. And most animals communicate with discrete signals.
>
> \>>> Why use a multi-round bargaining game to model language evolution by contrast to, say, an established signaling game-style setup with conflict of interests?
>
> If we modify an established signaling game-style setup with conflict of interests, such as one’s goal is to signal others to a wrong direction and gain the reward, we do not see any motivation for the receiver to communicate in the first place. On the contrary, we have shown that agents can cooperate with a conflict of interest in the bargaining game.
>
> \>>> How is the "credit mechanism" and the "market" motivated/grounded in light of this being an investigation about language emergence?
>
> In short, the credit mechanism means the agent begins to realize that others would also tell lies, which is one step in the evolution process. Because no consensus is reached in the community and there is no external authority, the individual has no choice but resists this behavior on its own.
>
> As for market, it is just used in the bargaining game for a better understanding of how the game works. The core idea is to design a non-cooperative game with conflict of interests. We believe the non-cooperative game is extremely common in real-world and can be a representative setting.

---

> > ### Comment · AnonReviewer4 · 2020-11-16
> > **Clarifications**
> >
> > I'm sorry but some points are still unclear to me.
> >
> > Most importantly, there is a claim I saw repeated in different flavors throughout the answers to all reviewers. In answer to Reviewer 4:
> >
> > > [...] human languages, like species, evolve over time. As a complex communication system, language should evolve from an immature communication system at an early stage. This is the period we want to investigate when naïve communication protocol, not necessarily possessing the properties owned by language (e.g. cryptophasia created by twins is not culturally transmitted), was just created and used by agents in a community.
> >
> >
> > As an answer to Reviewer 1:
> >
> > > just as Rome was not built in a day, language evolved from naïve communication.
> >
> > And to Reviewer 3:
> >
> > > We argue that the emergence of language is a process of continuous improvement, in other words, we have simpler communication protocol, such as transmitting a few symbols representing proposals, at the very early stage. And it cannot be considered as language.
> >
> > These assumptions need be substantiated. They should minimally be contextualized in the literature from which they draw. I'm still not even sure whether we are still looking at language (answer to reviewer 1 and the title of the paper) or not (answers to reviewer 3 and 4)? Why is this naïve protocol of the right simplicity vs. alternatives? No animal signaling system is, as far as we can tell, quite like natural language, so why is Vervets' behavior relevant but not that of marmots or Zebra Finches?
> >
> > To be clear: I do agree with some of the authors' intuitions. However, these answers add strength to my impression that this manuscript would greatly benefit from a more fleshed out scientific and conceptual backbone.
> >
> > To my question
> > >> Is there a reason to use a discrete dynamic over a continuous one?
> >
> > You answer
> > >We use discrete symbols over continuous ones to communicate since it can be easily interpreted and analyzed. And most animals communicate with discrete signals.
> >
> > I'm afraid that's a misunderstanding. I wasn't asking about the signals. I'm asking about the [evolutionary] dynamic (e.g., the difference between a discrete-time replicator dynamic and a continous-time one).

---

> > > ### Author Response · Authors · 2020-11-17
> > > **Reply to AnonReviewer 4**
> > >
> > > For the argument that language evolves from naïve communication, some works make an agreement that prior to the emergence of language some PRE-ADAPTATIONS occurred in the hominid lineage (Morten H, 2003. Language evolution: consensus and controversies). In more detail, many works argue that one candidate is the ability to use symbols. For example, (Bickerton, D, 2003. Symbol and structure: a comprehensive framework for language evolution.) in the evolution of our species, the symbolism may have preceded syntax by as much as two million ago, besides, three features (modality, symbolism, and structure) of language evolved separately. (Iain Davidson, 2003. The Archaeological Evidence of Language Origins: States of Art; Deacon, 2003. Universal grammar and semiotic constraints.) also stress the work of understanding the relations between the earliest communication using symbols and the final stage of the evolutionary emergence of language.
> > >
> > > As (Deacon, 1997. The Symbolic Species-The Co-Evolution of Language and the Brain) points out, if we are interested in the origin of language, we need to understand the emergence of symbolic signals (naïve communication). Based on all these arguments, our investigating naïve communication protocol uses symbols to transmit meaning based on a social convention or implicit agreement.
> > >
> > > According to (Li and Hombert, 2002. On the evolutionary origin of language; Givón, 2002. The evolution of language out of pre-language.), there is a criterion separating pre-language communication from post-language communication in hominid evolution is the difference between linguistic change and the evolutionary change of communication in the animal kingdom. The pre-language communicative behavior of hominids, like animal communicative behavior, was subject to the constraints of Darwinian evolution. Those hominids who made the change achieved a higher level of fitness than those hominids who failed to make the change. While linguistic change, which began in the post-language communicative era of hominid evolution, however, is by and large tied to society and culture. Thus, we disregard the factors related to linguistic change since we are investigating pre-language communication in hominid evolution.
> > >
> > > As for lying, vervets’ lying is just one example to illustrate language is not the prerequisite of lying since no animal signaling system is like natural language. Nobody can ever find evidence about whether hominid would lie or not. However, there is no doubt human is more intelligent than other animals. Since so many animals show lying behaviors (Whiten, A. & Byrne, R. W., 1988. Tactical deception in primates; W Abberley, 2015. Animal Cunning: Deceptive Nature and Truthful Science in Charles Kingsley's Natural Theology), we believe it is a reasonable assumption that lying is possible at the early stage of hominid evolution.
> > >
> > > All in all, we argue that lying is possible in pre-language communication from which language evolved, and we are investigating what resists lying from undermining the motivation of communication using symbols.
> > >
> > > \>>>Is there a reason to use a discrete dynamic over a continuous one?
> > >
> > > If you are referring to why choosing K tournaments instead of one tournament before natural selection:
> > > Credit is meaningless when there is only one tournament since it varies based on the profits obtained from previous tournaments. In addition, global lying behavior only makes sense when the number tournament is large since the agent tells lies not only about details but also relevant to the global situation.
> > >
> > > If you are referring to why choosing K tournaments, not some continuous setting:
> > > Since we use playing games representing the competitions, corresponding one game with one discrete step is reasonable. The discrete step also allows us to apply reinforcement learning (dealing with problems in Markov decision processes) investigating how global lying deals with the credit mechanism in the second setting.

---

### Official Review · AnonReviewer1 · 2020-10-27
**Interesting, but needs clearer motivation and detail**

**Rating:** 5
**Confidence:** 3

**Review:**

# Overall review

This paper attempts to address a question in the emergent communication literature: what preserves / maintains the stability of emerged communication protocols.  The authors manipulate the prevalence of lying behavior in a community of agents playing a variant of a Nash bargaining game.  The main take-away is that explicit punishment, from the environment and from truth-tellers not wanting to communicate with liars, can prevent the spread of exploitative lying behavior in the community.

My main worry about the paper is that the conceptual motivation is a bit unclear.  The authors present the paper as addressing when emergent communication can be stable.  And, after identifying a condition under which a penalty for lying removes liars from the population, they conclude that "Only under such a environment, language is finally possible."  But in all experiments, the communication protocol is fixed: mappings from market needs to proposals.  In other words, what changes is less the communication protocol itself, and more the prevalence of lying / truth-telling.  It thus is not clear how these results are to be interpreted as being about the emergence of communication itself, and not alternative uses of a pre-existing protocol.

Pros:
* Addresses a foundational question in emergent communication.
* Explicitly models a community of agents, instead of just dyads.
* Identifies explicit factors influencing their outcome of interest.

Cons:
* Conceptual motivation a bit unclear (see above).
* Communication protocol is fixed, not emergent.
* Many interesting results / sub-experiments are presented as parentheticals / foot-notes, without full details in appendices.


# Small comments

* p 3, choice to not do simultaneous proposals.  More details about these experiments should probably be provided in an appendix.

* Natural selection mechanism: the authors remove the lowest-performing single agent.  This is relatively different from the standard interpretation, which is that the number of offspring in the next generation is proportional to an agent's fitness.  Did the authors experiment with that approach, or can they say more to motivate theirs?

* The y-axis in the main results are "average scaled reward".  How exactly is the scaling done?  This would help in interpreting the results, as would comparison with a random baseline and/or an optimal strategy.

* "As the natural selection approaches, there is less and less need to pretend for liars ..."  I had some difficulty following the reasoning here.  Are the 10 tournaments _before_ a single episode of natural selection?  Or did natural selection occur in the middle?  If the former, it would help to know if these are averaged across "generations" in order to support the authors' interpretation.


# Minor typographic comments

* abstract: "From the perspective of Darwinian, ..." needs to be finished after "Darwinian".  Same at the end of page 1.

* p 1, "some intrigue properties" should be "some intriguing properties"

* p 2, "lairs" should be "liars"; this one recurs a few times in the text.

* p 8, "artificial intelligent" should be "artificial intelligence"

---

> ### Author Response · Authors · 2020-11-15
> **Reply to AnonReviewer 1**
>
> \>>> My main worry about the paper is that the conceptual motivation is a bit unclear…
>
> Language is a structured system of communication. However, just as Rome was not built in a day, language evolved from naïve communication. But lying behaviors emerge as long as communication is possible. As normally assumed, once a naïve communication protocol has been created, it is more convenient and efficient for agents to cooperate and gain more profits, thus they have more intention and motivation to develop a more complex communication protocol. However, what if, at this stage, some agents happen to learn how to lie? We have shown that, once lying behavior appears, liars would be dominant in the community, leading to less need and motivation for cooperation. In addition, the motivation for communication would also perish. Therefore, in our setting it is unnecessary to investigate the change of protocol itself, not to mention, existing naïve communication protocol may also die out according to Lamarckism. In our settings, the naïve communication protocol mentioned above is transmitting a few symbols representing proposals.
>
> \>>> choice to not do simultaneous proposals…
>
> Since first-mover has been mitigated by the strategy in (Cao, ICLR 2018. EMERGENT COMMUNICATION THROUGH NEGOTIATION), there should be no difference between these two kinds of non-cooperative games. Moreover, empirically agents cannot make use of communication to obtain extra profits in the simultaneous setting. In more detail, one party does not know what the other would offer, which incurs additional mistrust. Therefore, their proposal would be more conservative by giving an evenly dividing proposal. Then communication is useless since nothing valuable can be inferred.
>
> \>>> Natural selection mechanism
>
> In our setting, the number of offspring is not proportional to fitness, but it still follows the core principle. We deterministically remove the offspring of the weakest agent and add one more offspring for the strongest agent. The main purpose is to accelerate evolution.
>
>
> \>>> The y-axis in the main results are "average scaled reward"…
>
> The scaled reward is the rewards normalized by the maximum reward achievable for each agent, and scaled reward shows the relationship with the optimal strategy of each agent. We did experiments about random policy and the result shows it is around 0.5, which is similar to the evenly dividing strategy.
> .
> \>>> "As the natural selection approaches, there is less and less need to pretend for liars ..."
>
> After 10 tournaments, natural selection happens. As natural selection approaches, it means they are retiring from the stage of history. Therefore, if the liar still has credits, it is desired to use them to get more profits before natural selection, otherwise, it would be wasteful.

---

### Official Review · AnonReviewer3 · 2020-10-28
**Interesting Setting but Implications of Results are Unclear or Speculative**

**Rating:** 6
**Confidence:** 3

**Review:**

**Summary:** This paper studies multi-agent communication with an aim to mimic conditions for the emergence of language in society. They argue that the emergence of language requires co-operation. However, people can cheat and make a profit by lying, and eventually, people have no incentive to co-operate.

A proxy multi-agent setting is created to study this which proceeds in a series of interaction between two (randomly sampled) agents. In each interaction, there is a market demand for N items, and both agents have a fixed quantity of these items and their own hidden utility. An agent makes a proposal to another agent for a set of goods which consists of quantities of each item that they are willing to give. The other agent has to fulfil the remaining quantity to match the market demand. The other agent can accept the proposal at which point both agents get reward based on the remaining items and their respective utilities. Otherwise, the bargaining will continue and the other agent will make a proposal. There is a timeout after which both get a reward of 0.  There are certain agents who are chosen as "liars" who can make a proposal but refused to follow through leading to the remaining market gap being split equally.  Lying can, therefore, result in profit. A credit mechanism is also introduced to measure the performance of agents. Liars can use credit scores to decide whether to lie or not. A neural network is used to generate proposals and REINFORCE style training is performed to optimize reward. Authors also study "natural selection" where low-ranking agents get eliminated after a certain time, and "genetic evolution" where high-ranking agents produce clones.

Three different settings are studied: (i) where there are no measures taken to stop liars, (ii) where each truth-teller can refuse to play with an agent with negative credit score, (iii) where global measures are taken including reward penalty for lying or liar agents being banned for a fixed time. It is found that in both (i) and (ii); liars take over society. Individual counter-measures slow down the proliferation of liars but don't stop them. However, global measures in (iii) are able to stop liars. Authors argue that global measures in human society (reciprocal altruism) is the reason why conditions were right for language to evolve.

**Strength:**
- The setting is interesting. In general, understanding the dynamics of language evolution is a very interesting topic. This can shed light on human history, and maybe potentially be helpful for developing language understanding agents.

**Weakness:**
- The language part of the story is confusing since there is no language used in the experiments. In fact, what is really being asked here is if we can have a society without cooperation? Since, if everyone ends up being a liar then we don't have a society. Therefore, the language part feels like a red-herring.

- Implications of the results are unclear as it is not obvious that the setup captures all the important nuances present in human society. E.g., in a real-world, people who are speaking the truth can help each other by sharing technology that can give them a survivor edge over liars. This happens even if liars are not directly punished for their deeds. Then there is the question if the way natural selection and genetic evolution are modelled, reflect the real-world. I am not an expert on evolution but the paper doesn't say enough to justify the choice.

**Questions:**

- In the case where agent "i" lies, the other agent will still end up paying its share of the accepted proposal plus 50% of what agent "i" promised. Is this right?

- Can one pay more items than they possess? If not, do we always have $d_n \ge q^i_n + q^j_n$ for every $i$ and $j$, as otherwise there are not enough items to satisfy the market?

- In the setting of individual resistance, once an agent ends up with a negative credit score then the truth-tellers will refuse to play with them resulting in an addition of $1 - \sqrt{e} < 0$ to their credit score. So can an agent never increase their total reward once their credit score becomes negative? Or, perhaps liars will still play with them and so there is a chance for their total reward to increase.

- How would the behaviour be if the agent also had access to (i) total reward earned by the other agent, (ii) percentage of cases the other agent lied. E.g., if an agent saw that the other agent lie then they are likely to either not play with them or demand a more lucrative offer before committing. This can allow for more interesting policies to evolve.

---

> ### Author Response · Authors · 2020-11-15
> **Reply to AnonReviewer 3**
>
> \>>>The language part of the story is confusing since there is no language used in the experiments. In fact, what is really being asked here is if we can have a society without cooperation? Since, if everyone ends up being a liar then we don't have a society. Therefore, the language part feels like a red-herring.
>
> We argue that the emergence of language is a process of continuous improvement, in other words, we have simpler communication protocol, such as transmitting a few symbols representing proposals, at the very early stage. And it cannot be considered as language. Then communication protocol has continuously improved more and more sophisticatedly and language is finally possible. Besides, lying behavior is possible immediately when agents are capable of communicating with each other. Therefore, what we want to express is: agents somehow master a naïve communication protocol at a first stage, then once lying behavior appears in the community, it is unlikely to further develop a more sophisticated communication protocol among agents since lying undermines the motivation of cooperation as well as communication.
>
> \>>> Implications of the results are unclear as it is not obvious that the setup captures all the important nuances present in human society…
>
> In the second experiment settings, it is worth noting we are investigating a community that is at the early stage and extremely unstable. The agents do not have enough cognition to understand obligatory reciprocal altruism, on the contrary, they are rather short-sighted driven by selfish nature. Sharing information is an altruistic act and should not occur according to standard darwinian theory (Ib Ulbaek, 1998. The origin of language and cognition). Evolution demands that we enhance our fitness, not others. And the question arises, why should agents help others without any profits?
>
> In the third setting where obligatory reciprocal altruism has been evolved, social pressure dominates since it influences agents more severely and regulates agents more effectively than other factors (including sharing information) due to its mandatory.
>
> Survival of the fittest is a way of describing the mechanism of natural selection originated from darwinian evolutionary theory. Our evolution mechanism obeys this core thought. Therefore, the weakest agent does not have offspring while the strongest agent has more offspring than other agents.
>
> \>>> In the case where agent "i" lies…
>
> Yes, the market will evenly and compulsorily collect the missing from both parties. Such a setting could mimic the scenario where lying can benefit oneself and damage the other party.
>
> \>>> Can one pay for more items than they possess?
>
> It is not allowed to pay for more items than they possess. Therefore, we always have $d_n \leq q_n^i + q_n^j$.
>
> \>>> In the setting of individual resistance…
>
> The liar with all negative credit scores to all the truth tellers would have no chance to play games with them. However, its reward could still increase since it can play with other liars.
>
> \>>> How would the behavior be if the agent also had access to…
>
> The main difference between individual pressure and social pressure is that social organizations have the right to access more information than the individual therefore it can influence liars more precisely. It is unrealistic for agents to have a God-like view to know others’ information. Besides, as we mentioned before, agents may not have the intention or motivation to share information with others. How to gradually evolve from individual pressure to social pressure is not the focus of this paper, but will be investigated in future work.

---

### Author Response · Authors · 2020-11-16
**We have posted the responses to each reviewer and we will continuously address any further comments.**

We look forward to fully discussing the comments and concerns of the reviewers. Hope they could fully understand the merits of the paper.

---

> ### Author Response · Authors · 2020-11-18
> **We would like to encourage reviewer 1 and 3 to respond to our responses.**
>
> Please let us know whether our responses have addressed your comments and concerns. Also let us know if you have further comments. We think this would greatly help us to improve the paper.

---

> > ### Author Response · Authors · 2020-11-24
> > **Summary of Response to Reviews**
> >
> > We thank all the reviewers for their thoughtful comments and suggestions.
> >
> > We have uploaded a revised version of the manuscript which addresses the concerns shared by the reviewers. In particular, we supplemented some motivations behind fundamental assumptions made in this work. In more detail, we explained what motivated us to investigate, the reasons why we disregarded linguistic change and the relationship between our work and the emergence of language. Secondly, we cleaned-up some sloppy terminology, corrected the typos, and revised Figure 3.

---

### Decision · Program_Chairs · 2021-01-07
**Final Decision**

**Decision:**

Reject

**Comment:**

The authors present a study on what maintains the stability of emerged communication protocols. To study this question the authors design experiments in bargaining communities of agents in 3 setups,  a) no punishment of restriction of liar agents b) allowing individual agents to refuse bargaining with  liar agents and c) introducing a global punishment system for liar agents.

Overall the reviewers agree that the design of the study is interesting, but also point that motivation and take-home messages of this study are unclear. Having read the paper, I share the same opinion. The authors discuss on a very abstract level about the implications of this study for the field of AI, but this study is quite specific and clearly does not capture all the complexities or real societies. From the scale of results and study, I think it would be more valuable to draw some concrete proposals/implications about perhaps multi-agent modelling or environment design in general.

All in all, this is an interesting study but some more work needs to be done around research framing.